# Plasma Polyamine Biomarker Panels: Agmatine in Support of Prostate Cancer Diagnosis

**DOI:** 10.3390/biom12040514

**Published:** 2022-03-29

**Authors:** Donatella Coradduzza, Tatiana Solinas, Emanuela Azara, Nicola Culeddu, Sara Cruciani, Angelo Zinellu, Serenella Medici, Margherita Maioli, Massimo Madonia, Ciriaco Carru

**Affiliations:** 1Department of Biomedical Sciences, University of Sassari, 07100 Sassari, Italy; donatella.coradduzza@libero.it (D.C.); sara.cruciani@outlook.com (S.C.); azinellu@uniss.it (A.Z.); mmaioli@uniss.it (M.M.); 2Department of Clinical and Experimental Medicine, Urologic Clinic, University of Sassari, 07100 Sassari, Italy; tatiana.solinas@aousassari.it (T.S.); madonia@uniss.it (M.M.); 3Institute of Biomolecular Chemistry, National Research Council, 07100 Sassari, Italy; emanuelagigliola.azara@cnr.it (E.A.); nculeddu@icb.cnr.it (N.C.); 4Department of Chemistry and Pharmacy, University of Sassari, 07100 Sassari, Italy; sere@uniss.it; 5Department of Biomedical Sciences and University Hospital of Sassari (AOU), 07100 Sassari, Italy

**Keywords:** prostate cancer, biomarkers, agmatine, LC-HRMS

## Abstract

Prostate cancer is the most frequent malignant tumour among males (19%), often clinically silent and of difficult prognosis. Although several studies have highlighted the diagnostic and prognostic role of circulating biomarkers, such as PSA, their measurement does not necessarily allow the detection of the disease. Within this context, many authors suggest that the evaluation of circulating polyamines could represent a valuable tool, although several analytical problems still counteract their clinical practice. In particular, agmatine seems particularly intriguing, being a potential inhibitor of polyamines commonly derived from arginine. The aim of the present work was to evaluate the potential role of agmatine as a suitable biomarker for the identification of different classes of patients with prostate cancer (PC). For this reason, three groups of human patients—benign prostatic hyperplasia (BPH), precancerous lesion (PL), and prostate cancer (PC)—were recruited from a cohort of patients with suspected prostate cancer (*n* = 170), and obtained plasma was tested using the LC-HRMS method. Statistics on the receiver operating characteristics curve (ROC), and multivariate analysis were used to examine the predictive value of markers for discrimination among the three patient groups. Statistical analysis models revealed good discrimination using polyamine levels to distinguish the three classes of patients. AUC above 0.8, sensitivity ranging from 67% to 89%, specificity ranging from 74% to 89% and accuracy from 73% to 86%, considering the validation set, were achieved. Agmatine plasma levels were measured in PC (39.9 ± 12.06 ng/mL), BPH (77.62 ± 15.05 ng/mL), and PL (53.31 ± 15.27 ng/mL) patients. ROC analysis of the agmatine panel showed an AUC of 0.959 and *p* ≤ 0.001. These results could represent a future tool able to discriminate patients belonging to the three different clinical groups.

## 1. Introduction

Prostate cancer is the most frequent neoplasm among men, and is the fifth leading cause of death worldwide, with the risk increasing with age [1]. Most cancers arise in the periphery of the prostate gland, and cause symptoms only when they have grown to compress the urethra or invade the sphincter or neurovascular bundle [2]. Early prostate cancer usually does not cause pain, and most affected men exhibit no noticeable symptoms. Within this context, clinical assessment of the onset of this disease is complicated by its clinically silent and biologically non-aggressive form. Severities and outcomes of prostate cancer is differ widely. Early-stage prostate cancer can usually be treated successfully, while some older men have prostate tumours that grow so slowly, being biologically non-aggressive, that they may never cause health problems during their lifetime. On the other hand, in other men, the cancer is much more aggressive. Some cancerous tumours can invade surrounding tissue and spread to other parts of the body, leading to metastatic cancers. PSA screening is recommended, by the European Society of Medical Oncology (ESMO) after the age of 50 [3]. Screening tests for prostate cancer include digital rectal examination (DRE), prostate-specific antigen (PSA) blood test, trans rectal ultrasound guided (TRUS) biopsy and prostate biopsy, with the latter being considered the gold standard [4]. Screening and regular follow-up, in which PSA determination plays an important role, is an important step in reducing cancer mortality. Nevertheless, there is still a large grey area of patients that have threshold values for this marker, making it confusing for diagnosis [5,6]. Actually, this leads to overdiagnosis, increasing the burden on the health care system with the unnecessary implementation of invasive diagnostic and therapeutic interventions [7]. Serum and plasma tests are less invasive and faster to perform than tissue biopsies, thereby making them more suitable [8]. Recently, diagnostic tests based either on biomarkers found in urine, such as PCA3, T2-ERG, exosome, and others, or in blood, such as four kallikrein proteins, have been studied [5,6,9,10]. Within this context, different authors have proposed a multi-analyte blood test (Cancer-SEEK) for assessing cancer type based on multiple analytes [11,12]. Furthermore, since inflammation seems to play a key role in the evolution of cancer diseases [13], the role of several biomarkers of inflammation [14] derived from blood counts has been investigated: neutrophil/lymphocyte ratio (NLR), NLR derivative (dNLR = neutrophils/(white blood cells-neutrophils)), platelet/lymphocyte ratio (PLR), monocyte/lymphocyte ratio (MLR), (neutrophils × mono-cytes)/lymphocyte ratio (SIRI), and (neutrophils × monocytes × platelets)/lymphocyte ratio (AISI). Many authors consider the role of polyamines belonging to the arginine and lysine cycle to be very relevant among circulating biomarkers, and changes in their plasma concentration have been studied as biomarkers for various cancers.

Polyamines are small aliphatic polycations that bind several negatively charged molecules under physiological conditions, including DNA, RNA, ATP, certain types of proteins, and phospholipids [7,8]. Thus, they play an essential role in cell growth, proliferation, differentiation, development, immunity, migration, gene regulation, DNA stability, and protein and nucleic acid synthesis [4,7,9,10]. Other functions include cell adhesion and extracellular matrix repair, and they are involved in specific signalling processes. Polyamine levels are maintained within a narrow range, leading to severe physiological effects when dysregulated. Indeed, a pronounced decrease in polyamine levels can prevent cell proliferation and migration [9,15]. On the other hand, an excess of polyamines results in apoptosis and cell transformation [9]. In mammalian cells, the parental polyamines are synthesised from ornithine by an initial decarboxylation step to putrescine, catalysed by the enzyme ornithine decarboxylase (ODC) [16]. This step is followed by other enzyme-catalysed aminopropyl transfer reactions via spermidine synthase (SRM) and spermine synthase (SMS), generating spermidine and spermine, respectively [8,9,17].

Recently, in mammalian systems, agmatine was identified as a new polyamine from arginine decarboxylation [18,19].

Agmatine is an important biogenic amine, and could regulate polyamine metabolism [10], playing key roles in cellular metabolism and the inhibition or induction of cell proliferation, depending on its numerous interactions with the tumour environment [20,21,22,23,24,25,26,27,28,29,30]. Polyamines and their metabolites exist in both tissue and physiological fluid; their distribution throughout the body is not the same, especially in carcinogenesis [31,32]. ODC is the restricting enzyme, regulated by androgens in the prostate gland [33], and the gene encoding ODC is markedly induced in human prostate cancer [34]. It is well known that cancer cell metabolism is dependent on arginine, the so-called ‘Achilles’ heel’ of cancer, making it a potential target for cancer treatment [23,35,36,37]. Researchers and physicians have expressed the need for biomarkers that are able to discriminate patients in the early stages of prostate cancer (PC) from those with benign prostatic hyperplasia (BPH) or precancerous lesions (PL), the clinical evolution of which is not always predictable. Benign prostatic hyperplasia (BPH)—also called prostate gland enlargement—is a common condition as men get older. Having BPH does not increase a patient’s risk for prostate cancer. An enlarged prostate gland can cause uncomfortable urinary symptoms, such as blocking the flow of urine out of the bladder. The PL category includes patients with a histopathological diagnosis of atypical small acinar proliferation (ASAP) and premalignant proliferation of atypical ductal and acinar cells. Prostatic intraepithelial neoplasia (PIN), particularly high-grade PIN (HGPIN), and atypical small acinar proliferation (ASAP) have been identified as being precancerous lesions of the prostate, that is, they function as precursor lesions to prostatic carcinoma. These are categories with increased risk of developing prostate adenocarcinoma. The chance that a patient diagnosed with ASAP will develop cancer is about 40% [38]. The morphological appearance of HGPIN (i.e., tufting, micropapillary, cribriform, flat) is not always correlated with the consequent development of the neoplasm, meaning that clinical follow-up is highly recommended [39,40,41,42,43,44,45,46,47,48]. PIN refers to the precancerous end of a morphologic spectrum involving cellular proliferation within prostatic ducts, ductules, and acini. HGPIN possesses high predictive value for the future development of adenocarcinoma [49], and ASAP has potential significance for the development of synchronic malignant disease close to the source of origin of the biopsy [50]. Both are of great clinical importance for the early diagnosis of PCa. Whether or not the extent of high-grade PIN in biopsies is a predictor of prostate cancer is still controversial [51].

The aim of the present study was to investigate the levels of circulating polyamines in a population of subjects with suspected prostate cancer in order to understand whether they are capable of differentiation among different patient groups and could thus be used for clinical decision support.

To achieve this goal, high-resolution mass spectrometry (HRMS) was used in combination with high-performance liquid chromatography (HPLC), both of which are highly convenient due to their extraction of target ions from the total ion chromatogram, meaning that even minor polyamines can be detected and quantitatively analysed in complex matrices [52].

## 2. Materials and Methods

### 2.1. Chemicals and Methods

The reference standards of putrescine, cadaverine hydrochloride, spermidine hydrochloride, spermine, agmatine sulphate salt, *N*-acetyl-putrescine hydrochloride, *N*-acetylspermine trihydrochloride, *N*-acetylspermidine dihydrochloride, l-ornithine hydrochloride, lysine, l-arginine, aminobutyric acid, deuterated histamine, heptafluorobutyric acid (HFBA) and methanol were purchased from Sigma-Aldrich (St. Louis, MO, USA) for analytical type analysis. Water for LCMS was purchased from Fisher Scientific (Fair Lawn, NJ, USA).

Liquid chromatography–high-resolution mass spectrometry (LC-HRMS) analysis was performed using an UPLC Ultimate 3000 (Thermo Fisher-Dionex San Jose, CA, USA) system equipped with a HESI-II electrospray source to a Q-Exactive-Orbitrap™-based mass spectrometer (all from Thermo Scientific, San Jose, CA, USA). Chromatographic separation was carried out on the C18 column of the Gemini C18 (Phenomenex, Torrance, CA, USA), 100 mm × 2 mm, particle size 3 µm, the column was held at 37 °C. Chromatographic separation was achieved with gradient elution using a mobile phase composed of 0.05% heptafluorobutyric acid (HFBA) in water (A) and 0.05% HFBA in methanol (B).

### 2.2. Plasma Prostate Cancer Sample Preparation

A total of 170 male patients were recruited at the Urology Department of the University Hospital of Sassari between September 2018 and September 2019. The studies were conducted in accordance with the Declaration of Helsinki. Written informed consent was obtained from each subject before the study. Following the clinical diagnosis criteria (TRUS, PB, and biopsy analysis), three subsets of patients were ordered; 92 patients had prostate cancer diagnosed (PC), 50 had benign prostatic hypertrophy or no evidence of malignancy (BPH), and the remaining 28 had precancerous lesion (PL). Table 1 shows detailed information on demographic, pathological and treatment patients list of the clinical patient’s characteristics. All plasma samples were stored at −80 °C for one month between collection and measurement. An aliquot of 250 μL of plasma was transferred into an Eppendorf microtube and mixed with 150 μL of methanol (containing 0.05% HFBA) and 100 μL of water for 50 s. After precipitation, samples were centrifuged for 9 min at 15,000 rpm, and frozen overnight at −20 °C. The supernatant was evaporated to dryness at 36 °C under a stream of nitrogen. The residue was reconstituted into 500 μL of mobile phases and 50 μL of IS (internal standard, deuterated histamine). An aliquot of 5 μL of the solution was injected into the LC-HMRS system for analysis. Plasma sample preparation was performed in the same manner as the quality control (QC) samples. The supernatants obtained from these solutions were also used as the QC samples. The QC sample was a mixture of all samples, containing all information in the plasma samples, and it was used to optimise and supervise the injection process. QC samples were injected occasionally to test the stability of both the samples and the system during acquisition. Prior to sample analysis, the QC samples were injected six times to monitor the stability of the instrument. The six QC samples were then processed in parallel and injected to assess the repeatability of the method.

### 2.3. Arginine Decarboxylase (ADC) Quantification

The concentration of human arginine decarboxylase was evaluated using Human ADC (Arginine decarboxylase) ELISA Kit (Nordic Bioscience AB, Propellervägen, Sweden). Plate was washed twice before starting experiments according to the manufacturer’s instructions. After washing, 100 μL of each sample was incubated in a pre-treated plate at 37 °C for 90 min. After two washing steps in washing buffer, 100 μL Biotin-labelled antibody working solution was added in each well and incubated incubate at 37 °C for 60 min. Antibody was then removed and replaced by HRP-Streptavidin Conjugate solution for 30 min at 37 °C. After several washing steps, TMB liquid substrate was incubated at 37 °C in dark for 20 min and then stopped by the stop solution provided by the kit. Colour development was analysed at 450 nm using a microplate reader (Akribis Scientific, Common Farm, Frog Ln, Knutsford WA16 0JG, Great Britain). Standard curves were prepared according to manufacturer’s instructions. The relative O.D.450 of each sample was expressed as the (O.D.450 of each well) − (the O.D.450 of blank well). Each sample was assayed in duplicate, and values were expressed as the mean ± SD of 2 measures per sample.

### 2.4. Statistical Analysis

Results are expressed as an average value (mean ± DS). The distribution of variables was evaluated using the Kruskal–Wallis test and applied in order to compare the groups. The Kruskal–Wallis rank sum was employed to evaluate the distributions of each variance in the three groups under observation, assuming the value *p* < 0.05 as statistically significant. Statistical comparisons among the groups of parametric variables were evaluated using the unpaired Student t-test. The non-parametric continuous variables were compared with the case of normally distributed samples and with the median ± median absolute deviation (MAD) in the case of non-normal sample distribution. Correlations among variables were estimated using Pearson correlation. To verify the presence of associations among variables potentially involved in the development of the disease, a logistic regression analysis was performed. The analysis of the receiver operational characteristics curve (ROC) was used to test the ability of polyamines to predict prostate cancer. ROC curves were obtained by calculating the area under the curve (AUC). A supervised analysis was carried out by applying the orthogonal partial discriminant analysis of the minimum square (OPLS-DA), representing a rotation of the corresponding PLS-DA models and simplifying the information in a only one predictive component while maintaining the same predictive capacity [53]. To avoid model overfitting, the corresponding PLS-DA models were validated by 300-fold permutation tests. The prediction strength of the model was evaluated by “Leave out” analysis. Variable Importance Parameter (VIP) values were used to assess the overall contribution of each X variable to the model, summed over all components and weighted according to the Y variation accounted for by each component. The number of terms in the sum depends on the number of PLS-DA components found to be significant in distinguishing the classes. The Y-axis indicates the VIP scores corresponding to each variable on the X-axis. Bars indicate the factors with the highest VIP scores, and thus are the most contributory variables in class discrimination in the PLS-DA model. Comparison of univariate peaks was performed for the integrals of distinct metabolites using the non-parametric Mann–Whitney U analysis. The statistical analysis was carried out using Statgraphics Centurion XVII (v.17.2) software, MedCalc for Windows, version 15.4 64 bit (MedCalc Software, Ostend, Belgium), and SIMCA-P version 13.0, (Umetrics AB, Umea, Sweden) [54].

## 3. Results

### 3.1. Clinical Data

Table 1 shows the most important clinical information of the patients, including age, body mass index, lifestyle habits (smoking, alcohol), family history of prostate cancer, total serum PSA, serum PSA index, blood cell counts (WBC, RBC, HGB, PLT), leukocyte counts (Lymphocytes, Neutrophils, Monocytes), plasma inflammatory indices (MRL, NLR, PLR), and combined plasma inflammatory indices (SIRI, AISI), Charlson comorbidities, G6PD, IPSS (international prostatic symptoms score), IIEF (international index of erectile function), and TRUS (trans rectal ultrasound).

Average age of PC patients was 70 ± 7.86 years, PL patients 68 ± 7.87 and BPH patients 65 ± 8.17, showing that patients were aged from 57 to 77.

In the total sample, the most common pathological change in the prostate was prostate cancer in 55.10% of cases, followed by BPH in 29.34% of cases, precancerous conditions (atypical small acinar proliferation—ASAP—and high-grade prostatic intraepithelial neoplasia—HGPIN) in 15.57% of cases, and atrophic and inflammatory changes in the prostate in 14.1 % of cases. Based on the diagnosis, patients were divided into three groups. The age distribution of pathological changes in the prostate demonstrates that patients with prostate cancer were the oldest, with an average age of 70 years, followed by patients with precancerous conditions (HGPIN and ASAP) with 68 years, while the youngest were patients with BPH, with an average age of 65 years. Total PSA values were significantly different between PC patients and BPH. The analysis by Kruskal–Wallis test indicates that there is a significant difference, with increasing age, in Charlson and a decrease in TRUSS between prostate cancer and benign prostatic hyperplasia.

The correlation between PC patients and PL patients for PSA exhibits an increasing trend, while those for Charlson and IIEF exhibit a decreasing trend.

### 3.2. Polyamine Analysis

Table 2 shows the levels of plasma polyamines, and of the related amino acids (arginine, lysine) and metabolites (GABA) in the three groups. Patients with prostate cancer exhibited a significant increase in GABA, spermine, spermidine, putrescine, cadaverine and lysine levels as compared to Pl patients. An opposite trend could be observed for agmatine and acetyl-putrescine levels.

Values of agmatine and acetyl putrescine were significantly decreased, while GABA, spermine, spermidine, putrescine, cadaverine, lysine and ornithine levels were increased, in PC patients as compared to BPH. The analysis by Kruskal–Wallis test indicated that there was a significant difference in the observed changes (Figure 1).

### 3.3. Mono and Multivariate Analysis

In Figure 1, the multivariate analysis using the PLS-DA method (partial least squares discriminant analysis) shows good discrimination between the three groups of samples; in Figure 2, the OPLS-DA—orthogonal partial least squares discriminant analysis—reports the score plots derived from the LC-HRMS spectra of the plasma and the corresponding loading plots of the coefficients obtained from the three groups. The orthogonal matrix makes it possible to explore the ‘orthogonal’ components more easily and to fully understand data set. The characteristics of the OPLS method have been investigated for the purpose of discriminant analysis (OPLS-DA), demonstrating that class-orthogonal variation can be exploited to increase classification performance in those cases in which the individual classes exhibit divergence with respect to within-class variation [52].

The supervised analysis was performed by applying orthogonal partial least squares discriminant analysis (OPLS-DA), which implies a rotation of the corresponding PLS-DA models and simplifies the concentration of information into one predictive component while maintaining the same predictive ability [53]. We found that patients in the different groups were distributed in different areas, which indicated the significant difference in the metabolic mode between the group with PC (GREEN), the PL group (RED), and the BPH group (BLUE). Figure 1 and Figure 2 show the corresponding 2D predictive scoring plot against the first orthogonal component; the three groups tended to cluster naturally. Compared to the unsupervised principal component analysis, the supervised OPLS-DA makes it possible to obtain the variables’ influence on projection (VIP), highlighting the differences between the groups (Figure 3 and Figure 4).

VIP is a parameter used to calculate the cumulative measure of the influence of individual X-variables on the model. Variable influence on projection is applied in multivariate clinical data analysis to achieve improved diagnosis of process dynamics. Variable influence on projection (VIP) is commonly used to summarise the importance of the X-variables in multivariate models based on projections to latent structures, e.g., the PLS and OPLS methods. VIP values as great as or greater than 1 point to the most relevant variables, and generally, VIP values below 0.5 are considered to be irrelevant variables. Thus, VIP for OPLS with positive contribution scores corresponds to the metabolites contributing to class discrimination in the OPLS-DA model. The number of terms in the sum depends on the number of OPLS-DA components that are significant for distinguishing classes. The Y-axis shows the VIP scores corresponding to each variable on the X-axis. To avoid model overfitting, the corresponding OPLS-DA models were validated by 300-fold permutation tests, shown in Figure 5. The resulting regression lines showed an intercept of R2 at 0.0249 and an intercept of Q2 at −0.246, indicating the validity of the model. This difference was maintained when the paired analysis of the samples from three groups was split. Based on the OPLS-DA results, there was a statistically significant difference between the three classes of patients. The scoring plot of the predictive component showed no overlap.

The accuracy for the optimal cut-off of biomarkers predicting disease progression (combined with pathological variables) was determined using ROC curves (Figure 6) and by univariate analysis (Appendix A). ROC curves were used to quantify the predictive accuracy of the metabolites. We observed no significant differences between the three groups using the ROC curves calculated for ornithine, acetyl-spermidine, spermine and acetyl-spermine; increased levels of cadaverine were observed in PC patients, while there was no significant difference between BPH patients and PL. ROC curves of putrescine and spermidine are indeed able to discriminate between PC and PL, while acetyl-putrescine significantly decreases in PC groups (Appendix A). The predictive ability of PSA was analysed using the ROC curves. The area under the ROC curve and its 95% confidence intervals for PSA are shown in Appendix A. The area under the curve for PSA is 0.685. Comparison of PSA among the three groups revealed no statistically significant differences. Analysis of the agmatine panel shows an AUC of 0.959 and *p* ≤ 0.001 when differentiating between PC and BPH patients (Figure 6).

### 3.4. Arginine Decarboxylase (ADC) Quantification

The concentration of ADC, an enzyme responsible for catalysing the conversion of L-arginine to agmatine and carbon dioxide, was quantified in the plasma of the PC, PL and BPH patient groups. Figure 7 shows an increase in ADC concentration in PL and BPH patients compared to PC patients, confirming the trend in plasma agmatine levels.

## 4. Discussion

Among the metabolites tested, agmatine, in terms of sensitivity and specificity, was significantly different among the three groups. This increase was further confirmed by the observed increase in arginine decarboxylase activity levels in PL and BPH compared to PC (Figure 7) [55,56]. Different authors have demonstrated that agmatine is able to inhibit polyamine synthesis by increasing the expression of antizyme mediating the degradation of ornithine de-carboxylase and increasing the spermidine–spermine acetyl transferase activity [55,57,58,59,60]. On the other hand, decreased polyamine levels, observed by us in PL and BPH patients, could stabilise the transcripts (p53, TGF-β, JunD) of the genes inhibiting growth [59,60,61,62,63,64]. The observed agmatine level trend in plasma fit perfectly with previous results described by other authors in different kinds of tumour [65].

Agmatine is derived from arginine by the action of arginine decarboxylase. Many studies have shown that arginine is an important player in a variety of different biological processes, such as cell growth, and becomes a limiting factor in states of rapid turnover (e.g., neoplasms). Arginine deprivation therapy is being investigated as an adjuvant therapy for cancer; however, arginine is also required for the immune destruction of malignant cells [66,67]. Agmatine, with is a derivative of arginine that is irregularly distributed in organs and tissues, has a dual behaviour. On one hand, it interferes with polyamine synthesis by reducing the activity of ornithine decarboxylase (ODC), while on the other hand, it inhibits the uptake of polyamines [68,69]; this interference in turn leads to the suppression of tumour cell proliferation in vitro [57,70]. In this context, other authors have also shown that agmatine inhibits the proliferation of cancer cell lines in vivo by interfering with the polyamine pathway [71]. Our results support evidence of a promising plasma biomarker for prostate cancer patients that could be used in current medical practice to reduce unnecessary biopsies. Analysis of ROC curves showed that variations in agmatine concentrations, in terms of sensitivity and specificity, may be promising discriminators among the three groups of patients examined.

## 5. Conclusions

The data collected in this study and their statistical analysis suggest effective correlations between variations in indices of systemic inflammation and polyamine levels, al-lowing classification of the three categories of patients examined. The data obtained encourage us to further explore the use of promising biomarkers of disease progression in future clinical practice to differentiate patients into diagnostic classes and identify high-risk patients at an early stage. The results not only contribute to the understanding of prostate cancer, but also increase the knowledge of the complex interrelationships between polyamine metabolism and tumour cell proliferation, also in relation to the possible role played by the microenvironment surrounding the tumour.

## Figures and Tables

**Figure 1 biomolecules-12-00514-f001:**
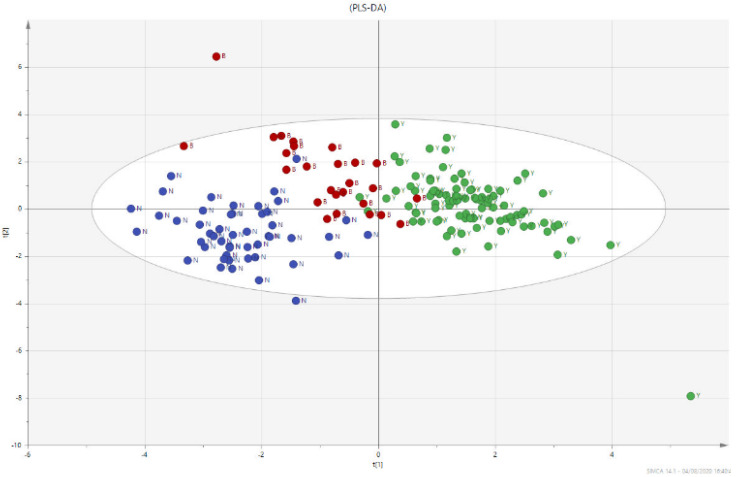
Multivariate analysis using PLS-DA method. PLS-DA loading plot shows a good dis-crimination of the three groups of samples, PC (GREEN), the PL group (RED) and the BPH group (BLUE).

**Figure 2 biomolecules-12-00514-f002:**
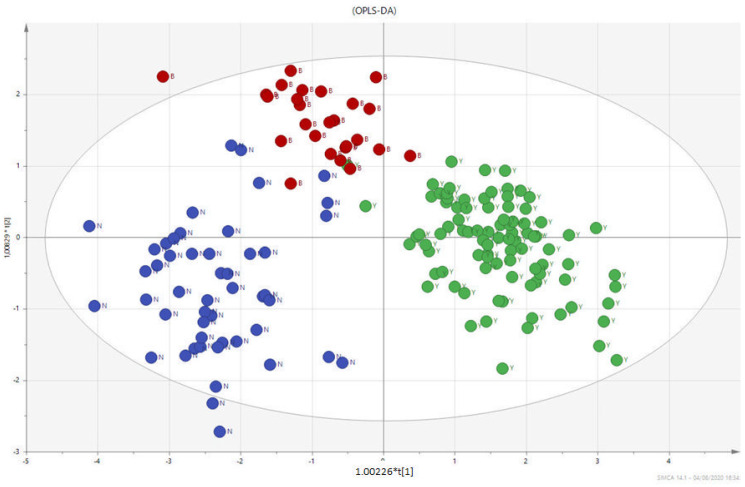
PLS-DA score plots derived from the LC-HRMS spectra of plasma and corresponding coefficient loading plots obtained from the three groups—PC (GREEN), PL (RED), and BPH (BLUE).

**Figure 3 biomolecules-12-00514-f003:**
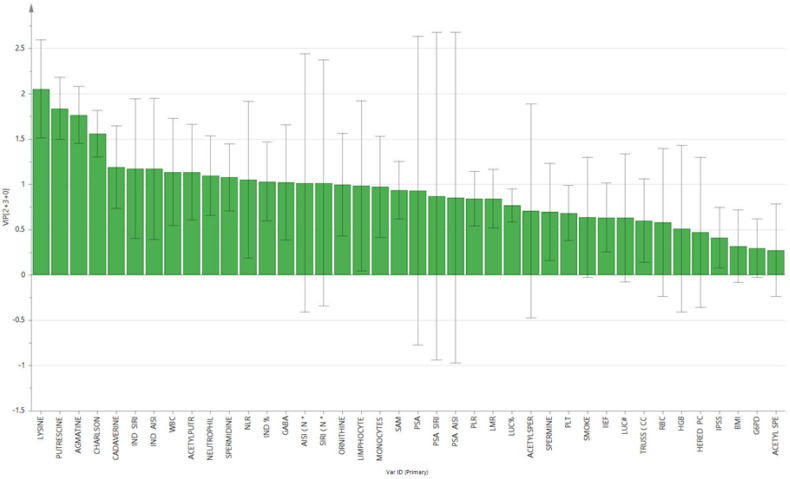
Contribution plot from model including total VIP; peaks with positive contribution scores correspond to metabolites with higher levels. The Y-axis indicates the VIP scores corresponding to each variable on the X-axis.

**Figure 4 biomolecules-12-00514-f004:**
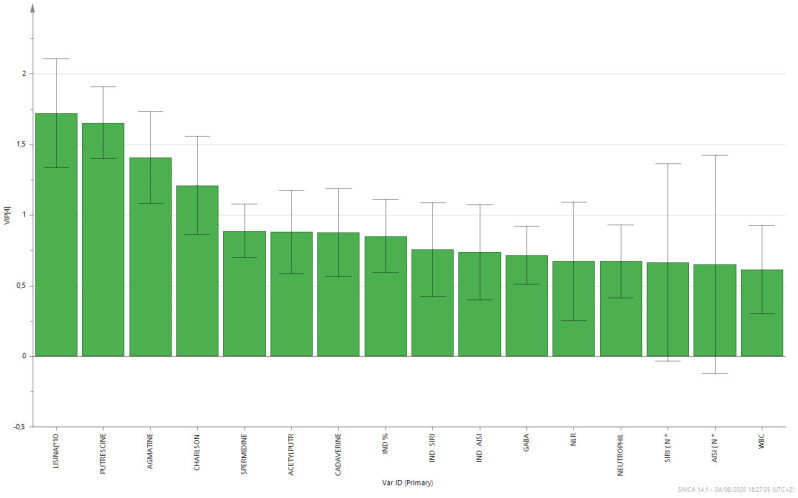
Factors with the highest VIP scores and contributory variables in class discrimination in the PLS-DA model.

**Figure 5 biomolecules-12-00514-f005:**
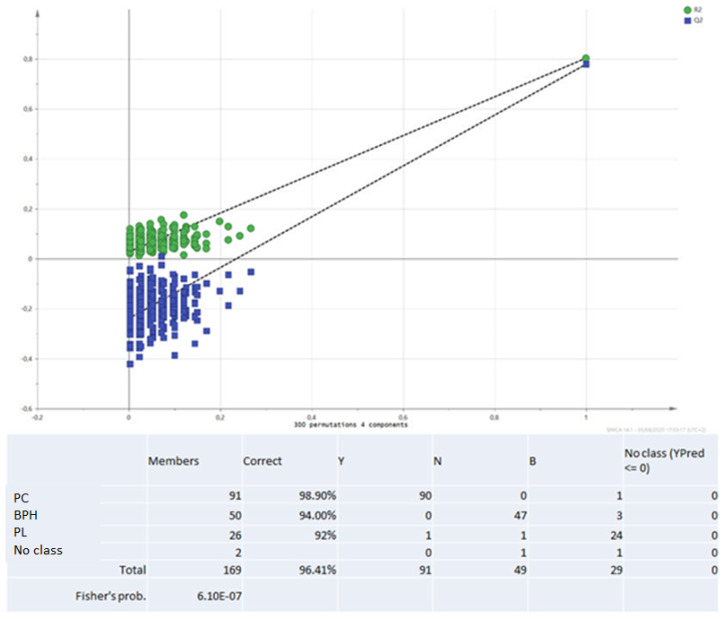
Validation method: 300-fold cross permutation validation plot. The Y-axis represents R2 (triangles) and Q2 (circles) for the model, and the X-axis designates the correlation coefficient between original and permuted response data.

**Figure 6 biomolecules-12-00514-f006:**
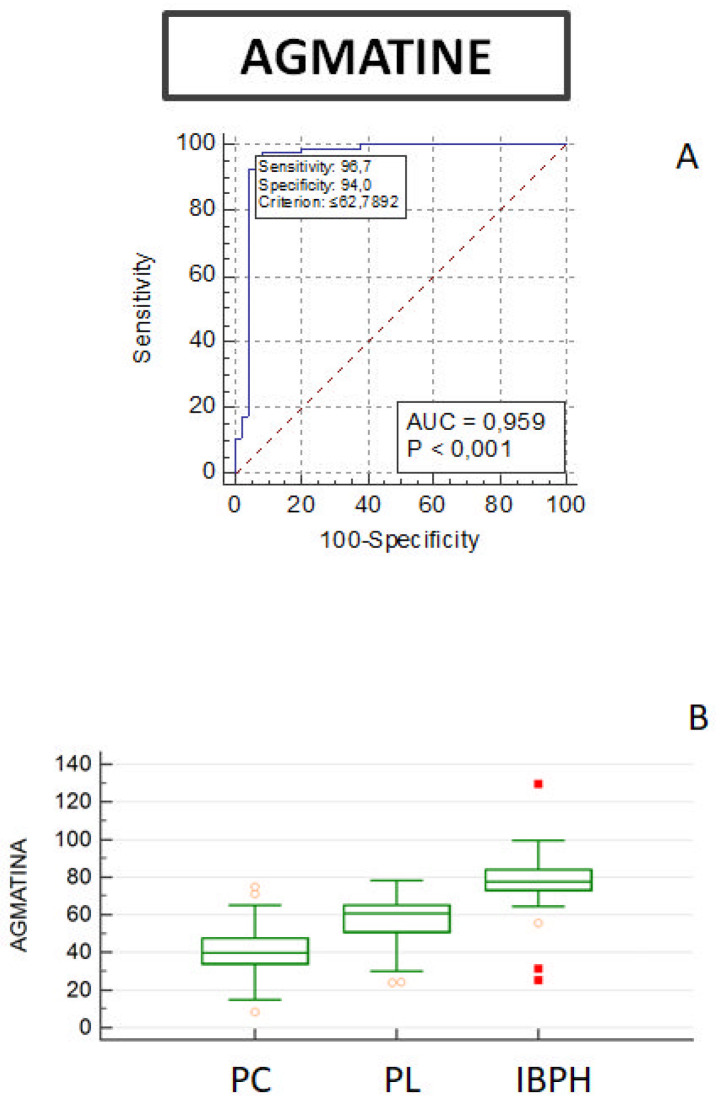
(**A**) ROC curve diagram of agmatine sensitivity and specificity in prediction of pathological conditions in PC and BPH patients. (**B**) Boxplots showing the distribution of agmatine among subjects with prostate cancer (PC, *n* = 92), precancerous lesions (PL, *n* = 26), or benign prostatic hyperplasia (BPH, *n* = 49). The centre line of the boxplots indicates the median (limits of the box indicate the 25th and 75th percentile). The whiskers represent either 1.5 times the interquartile range (IQR) or the maximum/minimum data point if they are within 1.5 times the IQR. Wilcoxon’s test was used to compare mean agmatine levels among groups.

**Figure 7 biomolecules-12-00514-f007:**
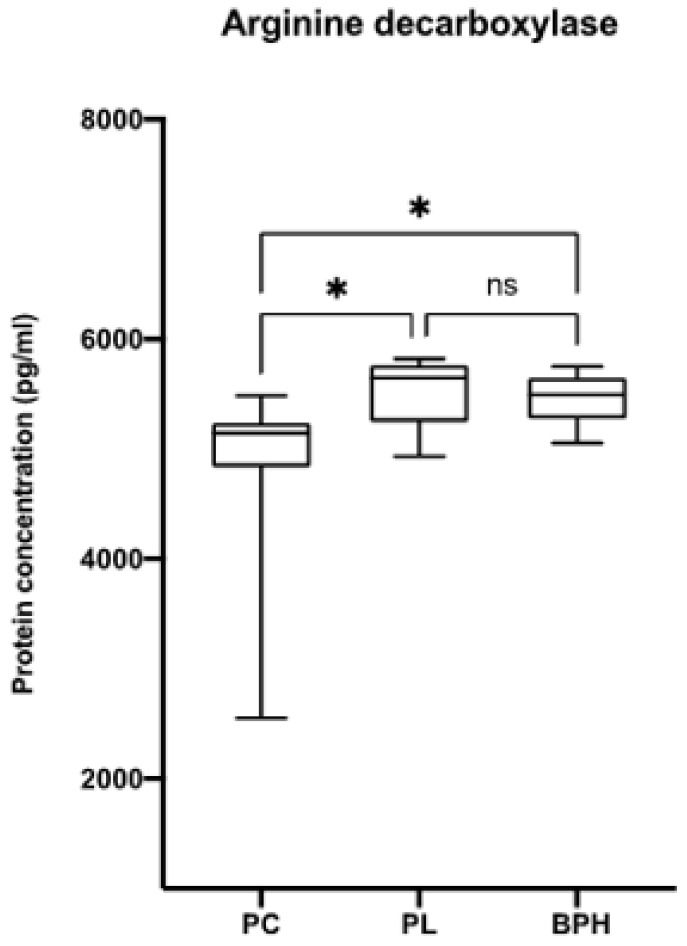
Arginine decarboxylase (ADC) quantification by ELISA. The concentration of ADC was measured in the plasma of patients from each group. Data are expressed as mean ± SD with reference to the control (mean ± SD; *n* = 170) (* *p* ≤ 0.05).

**Table 1 biomolecules-12-00514-t001:** List of characteristics from the clinical records of the patients. * PC vs. PL, ** PC vs. BPH. N.S. (not significant): denote a result from a statistical hypothesis-testing procedure that does not allow the researcher to conclude that differences in the data obtained for different samples are meaningful.

	PC = 92	PL = 26	BPH = 49	SIGNIFICANCE
AGE	70 ± 7.86	68 ± 7.87	65 ± 8.17	** *p* = 0.009
PSA	21.28 ± 45.09	6.38 ± 4.57	6.87 ± 6.80	** *p* = 0.0013
INDEX	12 ± 5.59	20 ± 11.11	19.48 ± 10.18	* *p* = 0.009, ** *p* = 0.007
WBC	7.73 ± 2.14	6.46 ± 1.45	7.33 ± 2.26	N.S.
RBC	5.07 ± 0.59	5.18 ± 0.93	5.24 ± 0.51	N.S.
HGB	14.20 ± 1.64	14.67 ± 2.16	14.75 ± 1.26	N.S.
RDW	13.99 ± 1.51	13.51 ± 0.95	13.60 ± 0.99	N.S.
HDW	2.64 ± 0.41	2.55 ± 0.35	2.52 ± 0.30	N.S.
PLT	235.5 ± 66.15	217.35 ± 45.01	235.60 ± 55.80	N.S.
NEUT	4.88 ± 1.89	3.96 ± 1.33	4.45 ± 1.91	N.S.
LYMPH	1.97 ± 0.79	1.77 ± 0.50	2.04 ±0.79	N.S.
MONO	0.50 ± 0.17	0.43 ± 0.13	0.47 ± 0.15	N.S.
EOS	0.20 ± 0.14	0.17 ± 0.10	0.23 ± 0.15	N.S.
BASO	0.04 ± 0.05	0.02 ± 0.04	0.04 ± 0.05	N.S.
LUC#	0.14 ± 0.07	0.12 ± 0.04	0.14 ± 0.06	N.S.
LUC%	1.96 ± 0.73	2.03 ± 0.57	2.13 ± 0.74	N.S.
LMR	4.16 ± 1.50	4.49 ± 1.80	4.47 ± 1.40	N.S.
NLR	2.92 ± 1.85	2.51 ± 1.67	2.47 ± 1.24	N.S.
PLR	137.69 ± 63.58	131.05 ± 42.02	132.45 ± 58.93	N.S.
SIRI	1.50 ± 1.28	1.15 ± 0.99	1.19 ± 0.94	N.S.
AISI	367.07 ± 338.04	247.24 ± 206.44	292.39 ± 289.63	N.S.
PSA/AISI%	0.10 ± 0.26	0.04 ± 0.03	0.04 ± 0.04	N.S.
INDEX/SIRI	11.75 ± 11.98	24.10 ± 29.58	12.49 ± 15.34	N.S.
INDEX/AISI%	0.06 ± 0.07	0.12 ± 0.16	0.06 ± 0.07	N.S.
FAMILIARITY	8/92 (8.69%)	6/26 (23.07%)	6/49 (12.24%)	N.S.
CHARLSON	5.22 ± 1.62	2.47 ± 1.19	2.75 ± 1.37	* *p* = 0.02, ** *p* = 0.013
G6PDH DEFICIT	7/92 (7.60%)	3/26 (11.53%)	5/49 (10.2%)	N.S.
BMI	27.40 ± 3.67	26.87 ± 2.82	26.80 ± 4.01	N.S.
IPSS	11.88 ± 6.43	12 ± 8.51	12.93 ± 9.47	N.S.
IIEF	13.06 ± 7.40	17.29 ± 6.17	15.75 ± 8.43	* *p* = 0.02
TRUS	51.26 ± 24.98	60.31 ± 33.05	65.45 ± 35.46	** *p* = 0.009
SMOKE	32/92 (34.78%)	4/26 (15.38%)	13/49 (26.53%)	N.S.
ALCOHOL	1/92 (1.09%)	0/26 (0%)	2/49 (4.08%)	N.S.

**Table 2 biomolecules-12-00514-t002:** Level of plasma polyamines, plasma correlated amino acids (arginine, lysine) and metabolites (GABA) (ng/mL). * PC vs. PL, ** PC vs. BPH, *** BPH vs. PL.

POLYAMINES	PC	PL	BPH	SIGNIFICANCE
AGMATINE	39.9 ± 12.06	55.31 ± 15.27	77.62 ± 15.05	* *p* = 0.007, ** *p* = 0.01, *** *p* = 0.009
GABA	30.03 ± 14.97	16.83 ± 12.54	22.02 ± 13.41	* *p* = 0.01, ** *p* = 0.008
SPERMINE	3.74 ± 2.20	2.8 ± 1.94	2.97 ± 1.76	* *p* = 0.01, ** *p* = 0.01
SPERMIDINE	8.43 ± 3.03	7.02 ± 1.78	5.31 ± 1.49	* *p* = 0.009, ** *p* = 0.007, *** *p* = 0.01
PUTRESCINE	14.28 ± 8.43	7.56 ± 1.62	6.45 ± 2.21	* *p* = 0.01, ** *p* = 0.01, *** *p* = 0.01
ACETYLPUTRESCINE	0.06 ± 0.04	0.14 ± 0.17	0.16 ± 0.10	* *p* = 0.008, ** *p* = 0.01
ACETYLSPERMINE	2.42 ± 0.77	2.68 ± 1.33	2.27 ± 0.49	*** *p* = 0.007
ACETYLSPERMIDINE	0.35 ± 0.24	0.4 ± 0.27	0.38 ± 0.24	N.S.
CADAVERINE	2.53 ± 0.81	1.75 ± 0.68	1.75 ± 0.67	* *p* = 0.01, ** *p* = 0.01
ARGININE	6.02 ± 2.30 × 10^4^	5.59 ± 1.87	5.39 ± 1.88	N.S.
LYSINE	2.33 ± 0.86 × 10^4^	1.64 ± 0.58	6.93 ± 2.06	* *p* = 0.006, ** *p* = 0.01
ORNITHINE	0.83 ± 0.29 × 10^4^	0.91 ± 0.17	1.04 ± 0.41	** *p* = 0.01

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
