# Peer review of "Plasma Polyamine Biomarker Panels: Agmatine in Support of Prostate Cancer Diagnosis"

_biomolecules, 2022, doi:10.3390/biom12040514_

Round 1

Reviewer 1 Report

In this manuscript, Donatella et al. are investigating the possibility of using the levels of circulation polyamines as biomarkers for differentiating between high- and low-risk patients with suspected prostate cancer. Based on the data presented in this study, it seems that the polyamines, agmatine in particular, are promising biomarkers for prostate cancer diagnosis. In my opinion, the study presents some interesting data and provides new insights into the metabolic pathway related to prostate cancer progression. However, there are still some aspects that need to be addressed before it can be published in Biomolecules:

  1. Has an un-targeted metabolomics analysis been performed with this study? If yes, it would be interesting to show if there are any metabolites other than polyamines with significant changes in different groups.
  2. Different statistical analysis methods have been applied to this study. But what metabolite profiles were used for each of these different analyses? Are all the analyses based on the polyamine profiles shown in Table 2, or any more profiles were included? If yes, please include a complete list of the metabolites used for statistical analysis.
  3. The study is focusing on the profiles of circulating polyamines, what is major source for these metabolites? Do we know any specific cell population may contribute the most to the production of these polyamines?
  4. The quantitation of ADC has been performed with plasma samples in this study, it makes more sense to do the same quantitation with prostate tissue samples. Considering the difficulty of access to human prostate tissue samples, performing this kind of study with animal prostate cancer model is recommended.
  5. Minor points:
    1. [61] line 122 should be [60];
    2. [76] is missing;
    3. 1 should be placed in the introduction section, not the results section. A bit more introduction about polyamine metabolism and its functions should be included in the introduction.
    4. No labels for the 2nd figure in Fig. 6;

Author Response

  1. Con questo studio è stata eseguita un'analisi metabolomica non mirata? In caso affermativo, sarebbe interessante mostrare se esistono metaboliti diversi dalle poliammine con cambiamenti significativi nei diversi gruppi.

Grazie mille per il tuo commento. Questo studio non ha un'analisi metabolomica non mirata. Grazie mille per il tuo suggerimento che ci esibiremo in un altro studio.

  1. Diversi metodi di analisi statistica sono stati applicati a questo studio. Ma quali profili metabolici sono stati utilizzati per ciascuna di queste diverse analisi? Tutte le analisi si basano sui profili di poliammina mostrati nella Tabella 2 o sono stati inclusi altri profili? In caso affermativo, includere un elenco completo dei metaboliti utilizzati per l'analisi statistica.

Grazie mille per la tua precisa osservazione e i preziosi commenti Tutte le analisi si basano sui profili di poliammina mostrati nella Tabella 2, altri profili non sono stati inclusi.

  1. Lo studio si sta concentrando sui profili delle poliammine circolanti, qual è la principale fonte di questi metaboliti? Sappiamo che una specifica popolazione cellulare può contribuire maggiormente alla produzione di queste poliammine?

Grazie mille per il tuo commento. Nelle cellule dei mammiferi esistono tre possibili fonti di poliammine: biosintesi dagli aminoacidi costituenti, assorbimento di poliammine preformate dalla dieta o integratori, poliammine preformate, dalla flora intestinale 1 .

La principale fonte di poliammine proviene dalle cellule tumorali 2 . Le poliammine sono alchilammine policationiche che si trovano comunemente in tutte le cellule viventi. la disregolazione della poliammina è stata trovata in una varietà di tumori. Ad esempio, il metabolismo delle poliammine svolge un ruolo chiave nel cancro al seno, influenzando la morte e la proliferazione cellulare. È stato ampiamente descritto che le poliammine facilitano le interazioni dei fattori di trascrizione, come i recettori degli estrogeni e il fattore nucleare kB, con il loro specifico elemento di risposta ed essendo anche coinvolte nella proliferazione di modelli ER-negativi e altamente invasivi di cellule tumorali. 3-5Le cellule del cancro alla prostata mantengono la secrezione di poliammine mentre proliferano, suggerendo la loro necessità di un alto livello di flusso metabolico delle poliammine. La via metabolica della poliammina è stata identificata come un potenziale bersaglio dei tumori della prostata 6-28

  1. La quantificazione dell'ADC è stata eseguita con campioni di plasma in questo studio, ha più senso eseguire la stessa quantificazione con campioni di tessuto prostatico. Considerata la difficoltà di accesso ai campioni di tessuto prostatico umano, si consiglia di eseguire questo tipo di studio con un modello di cancro alla prostata animale.

Grazie mille per il tuo commento. I test su siero e plasma sono meno invasivi e più veloci delle biopsie tissutali e sono quindi più adatti [7].

La quantificazione dell'ADC è stata eseguita con campioni di plasma in questo studio perché la nostra ricerca è principalmente focalizzata sull'identificazione di nuovi biomarcatori per la diagnosi e la stadiazione del PCa, valutandoli dal plasma (PHI, 4Kscore, S3M), dall'urina (PCA3, TMPRSS2 -ERG, TDRD1, DLX1, HOX6), che rappresentano un test rapido, non invasivo e fattibile.

Il livello di PSA superiore a 4 ng/mL è considerato sospetto per il PCa, sebbene i livelli compresi tra 4 e 10 ng/mL siano considerati in una "zona grigia". La diagnosi e la terapia basate solo sul PSA possono portare a sovradiagnosi e trattamento eccessivo di tumori che non sono clinicamente significativi 29 [14,15]. In questo studio, miriamo a identificare nuovi biomarcatori, rappresentati da poliammine circolanti, e le loro combinazioni con il livello di PSA per una migliore diagnosi precoce di PCa localizzato.

Inoltre è già noto che livelli più elevati di poliammine sono rilevabili nei tessuti del cancro alla prostata 30 .

  1. Punti minori:
    1. [61] la riga 122 dovrebbe essere [60];

Abbiamo modificato di conseguenza

    1. [76] manca;

Abbiamo modificato di conseguenza

    1. 1 dovrebbe essere inserito nella sezione introduttiva, non nella sezione dei risultati. Un po' più di introduzione sul metabolismo delle poliammine e le sue funzioni dovrebbe essere inclusa nell'introduzione.

L'abbiamo spostato nella sezione introduttiva.

    1. Nessuna etichetta per la 2a  figura in Fig. 6.

L'abbiamo presentato

Riferimento:

  1. Wallace HM: Le poliammine nella salute umana. Atti della Società di Nutrizione 1996, 55:419-431.
  2. Li J, Meng Y, Wu X, Sun Y: Poliammine e vie di segnalazione correlate nel cancro. Cancer Cell International 2020, 20:1-16.
  3. Tobias KE, Shor J, Kahana C: c-Myc e Max transregolano il promotore della decarbossilasi dell'ornitina del topo attraverso l'interazione con due motivi CACGTG a valle. Oncogene 1995, 11:1721-1727.
  4. Ignatenko NA, Babbar N, Mehta D, Casero Jr RA, Gerner EW: Soppressione del catabolismo delle poliammine da parte di Ki-ras attivati ​​nelle cellule di cancro del colon umano. Carcinogenesi molecolare: pubblicato in collaborazione con l'MD Anderson Cancer Center dell'Università del Texas 2004, 39:91-102.
  5. Poulin R, Casero R, Soulet D: Recenti progressi nella biologia molecolare del trasporto di poliammine metazoiche. Amminoacidi 2012, 42:711-723.
  6. Manni A, Grove R, Kunselman S, Aldaz M: Coinvolgimento della via della poliammina nella progressione del cancro al seno. Lettere sul cancro 1995, 92:49-57.
  7. Gupta S, Ahmad N, Marengo SR, MacLennan GT, Greenberg NM, Mukhtar H: Chemioprevenzione della carcinogenesi della prostata mediante α-difluorometilornitina nei topi TRAMP. Ricerca sul cancro 2000, 60:5125-5133.
  8. Gilmour SK: Poliammine e cancro della pelle non melanoma. Tossicologia e farmacologia applicata 2007, 224:249-256.
  9. Upp Jr JR, Saydjari R, Townsend Jr CM, Singh P, Barranco S, Thompson JC: Livelli di poliammina e recettori della gastrina nei tumori del colon. Annali di chirurgia 1988, 207:662.
  10. Shantz L, Levin V: Regolazione dell'ornitina decarbossilasi durante la trasformazione oncogenica: meccanismi e potenziale terapeutico. Amminoacidi 2007, 33:213-223.
  11. Pegg AE: Regolazione dell'ornitina decarbossilasi. Giornale di chimica biologica 2006, 281: 14529-14532.
  12. Ikeguchi Y, Bewley MC, Pegg AE: Aminopropiltransferasi: funzione, struttura e genetica. Giornale di biochimica 2006, 139:1-9.
  13. Wallace HM, Pegg AE: S-adenosilmetionina decarbossilasi. Saggi in biochimica 2009, 46:25-46.
  14. Casero RA, Murray Stewart T, Pegg AE: metabolismo della poliammina e cancro: trattamenti, sfide e opportunità. Recensioni sulla natura Cancro 2018, 18: 681-695.
  15. Wang X, Ying W, Dunlap KA, Lin G, Satterfield MC, Burghardt RC, Wu G, Bazer FW: Arginina decarbossilasi e agmatinasi: un percorso alternativo per la biosintesi de novo di poliammine per lo sviluppo di concetti di mammiferi. Biologia della riproduzione 2014, 90:84, 81-15.
  16. Wu HY, Chen SF, Hsieh JY, Chou F, Wang YH, Lin WT, Lee PY, Yu YJ, Lin LY, Lin TS: Basi strutturali della regolazione dell'omeostasi delle poliammine mediata da antizimi. Atti dell'Accademia Nazionale delle Scienze 2015, 112:11229-11234.
  17. Bae DH, Lane DJ, Jansson PJ, Richardson DR: La vecchia e la nuova biochimica delle poliammine. Biochimica et Biophysica Acta (BBA)-Soggetti generali 2018, 1862:2053-2068.
  18. Pegg AE: Spermidina/spermina-N 1-acetiltransferasi: un regolatore metabolico chiave. Giornale americano di fisiologia-endocrinologia e metabolismo 2008, 294: E995-E1010.
  19. Pegg AE: Tossicità delle poliammine e dei loro prodotti metabolici. Ricerca chimica in tossicologia 2013, 26:1782-1800.
  20. Belting M, Mani K, Jönsson M, Cheng F, Sandgren S, Jonsson S, Ding K, Delcros JG, Fransson L-Ak: Glypican-1 è un veicolo per l'assorbimento della poliammina nelle cellule di mammifero: un ruolo fondamentale per il nitrico derivato dal nitrosotiolo ossido. Journal of Biological Chemistry 2003, 278:47181-47189.
  21. Kahana C: La famiglia degli antizimi per la regolazione delle poliammine. Journal of Biological Chemistry 2018, 293:18730-18735.
  22. Uemura T, Stringer DE, Blohm-Mangone KA, Gerner EW: Il trasporto della poliammina è mediato dai meccanismi di trasporto del vettore endocitico e del soluto nel tratto gastrointestinale. Giornale americano di fisiologia-fisiologia gastrointestinale e del fegato 2010, 299: G517-G522.
  23. Uemura T, Yerushalmi HF, Tsaprailis G, Stringer DE, Pastorian KE, Hawel L, Byus CV, Gerner EW: Identificazione e caratterizzazione di un esportatore di diammina nelle cellule epiteliali del colon. Journal of Biological Chemistry 2008, 283:26428-26435.
  24. Moriyama Y, Hatano R, Moriyama S, Uehara S: trasportatore di poliammine vescicolari come un nuovo attore nella trasmissione chimica mediata da ammine. Biochimica et Biophysica Acta (BBA)-Biomembranes 2020, 1862:183208.
  25. van Veen S, Martin S, Van den Haute C, Benoy V, Lyons J, Vanhoutte R, Kahler JP, Decuypere JP, Gelders G, Lambie E: La carenza di ATP13A2 interrompe l'esportazione di poliammine lisosomiale. Natura 2020, 578:419-424.
  26. Cervelli M, Pietropaoli S, Signore F, Amendola R, Mariottini P: Metabolismo delle poliammine e cancro al seno: stato dell'arte e prospettive. Ricerca e trattamento del cancro al seno 2014, 148:233-248.
  27. Linsalata M, Orlando A, Russo F: Agenti farmacologici e dietetici per la chemioprevenzione del cancro del colon-retto: effetti sul metabolismo delle poliammine. Giornale internazionale di oncologia 2014, 45: 1802-1812.
  28. Affronti HC, Rowsam AM, Pellerite AJ, Rosario SR, Long MD, Jacobi JJ, Bianchi-Smiraglia A, Boerlin CS, Gillard BM, Karasik E: Upregulation farmacologico del catabolismo delle poliammine con inibizione del percorso di salvataggio della metionina come efficace terapia del cancro alla prostata. Comunicazioni sulla natura 2020, 11:1-15.
  29. Munteanu VC, Munteanu RA, Gulei D, Schitcu VH, Petrut B, Berindan Neagoe I, Achimas Cadariu P, Coman I: biomarcatori basati su PSA, tecniche immaginarie e test combinati per una migliore diagnosi del cancro alla prostata localizzato. Diagnostica 2020, 10:806.
  30. Peng Q, Wong CY-P, Cheuk IW-y, Teoh JY-C, Chiu PK-F, Ng CF: The Emerging Clinical Role of Spermine in Prostate Cancer. Giornale internazionale di scienze molecolari 2021, 22:4382.

Reviewer 2 Report

Brief Summary: The aim of the study by Coradduzza et al., was to assess the diagnostic value of circulating plasma polyamine metabolite expression for the identification of prostate cancer at early stages of the disease. The authors extracted blood plasma from 170 patients with confirmed prostate cancer, benign hyperplasia, or “borderline” disease. Samples were analyzed by liquid chromatography-high resolution mass spectrometry (LC-HRMS) to determine the expression levels of polyamines and assess their relevance to prostate cancer identification. They determine that the expression of metabolites involved in the arginine-lysine pathway, in particular the low agmatine polyamine levels, can distinguish patients with prostate cancer from ones with pre-cancerous lesions, suggesting that this method could be a promising alternative to prostate-specific antigen (PSA) screening for prostate cancer. Although this study is interesting and deals with a timely issue in prostate cancer, lacks in both experimental and conceptual aspects.

Strengths of the study: 

  • Analyses directly relevant to the clinical phenotypes of prostate cancer patients
  • Potential establishment of additional (to PSA screening) method for diagnostic prostate cancer biomarkers

Weaknesses of the study:

  • Definition of “borderline” disease is very unclear and maybe not particularly relevant
  • Lack of solid justification as to why plasma polyamine biomarker assessment is superior to PSA screening for diagnostic purposes
  • Lack of additional PSA – polyamine expression correlation analyses that would enhance the clinical relevance of the study
  • Results and Figures section is very hard to follow and requires clarifications

Comments:

Abstract: This section is overall well-written. Comments:

  • The authors should revise English language throughout the manuscript [minor]
  • It is unclear what “borderline” disease is and should be clarified in the text. Are there clinical guidelines that define such disease state in prostate cancer? What are the parameters? [major]

Introduction: This section presents background information on the prostate cancer screening, including PSA measurement and other diagnostic tests. Comments:

  • What does “clinically-silent” and “biologically non-aggressive” mean? Do the authors mean clinically undetectable disease? The terminology used is parts of the manuscript is confusing and requires clarification. [major]
  • In line 77: Arginine metabolism may have been suggested a therapeutically targetable vulnerability in prostate cancer but claims of “Achilles’ heel’, may be a bit farfetched without solid experimental evidence. [minor]
  • In line 84: The sentence does not make sense. Consider revising. [minor]
  • Benign prostatic hyperplasia (BPH) and pre-cancerous lesion, including prostatic intraepithelial neoplasia (PIN), are interchangeable terms. By which criteria did the authors distinguish one from the other? They claim the presence of three distinct histopathological groups and base all their analyses on these, but it is not clear how these subgroups were defined. [major]
  • What does ASAP stand for? [major]

Materials and Methods: The experimental procedures and study design are comprehensively explained apart from the precise and clear definition of the histopathological patient groups. In some cases the authors refer to them as PC, PL and IBPH (Table 1), and others as YES, BORDER and NO (Table 2). What is the difference between hypertrophy and heteroplasia? Why is the distinction between these two subgroups important? [major]

Results and Figures: The results and figures are severely lacking. Comments:

  • The authors do not describe their results. Rather they outline what each figure shows without any context for the reader. The manuscript would benefit from better organization and potentially subheading in the result section. [major]
  • Table 1: PSA levels in the PL group are lower than of IBPH, but still there is not significant difference compared to PC. Could the authors clarify as to why that may be? [minor]
  • Figure 1 is not very informative and may be redundant. The author’s effort to showcase metabolic pathways of the detected polyamine is appreciated but the presentation is lacking. [minor].
  • Figures 2 and 3 could potentially be merged or figure 2 omitted from the main text. They show the same point. Also, the font size is too small to read the figures. [minor]
  • In line 250: The authors jump straight to Figure 6, bypassing Figures 4 and 5. [major]
  • Figures 4 and 5 are extremely hard to read and is unclear what In the comparisons are and to what extent. Are the authors comparing different conditions, the expression of different polyamines? [major]
  • In Figure 6, what to the number 1, 2, 3 stand for? [minor]
  • Figure 8 is the most clearly presented piece of data in the manuscript. It shows that Arginine decarboxylase (ADC) expression is significantly different between PC and either PL or IBPH. This point is not discussed in the manuscript as it suggests that this assay may be as diagnostically powerful as the PSA screening (Table 1). [major]
  • Lack of additional PSA – polyamine expression correlation analyses that would enhance the clinical relevance of the study [major]

Discussion and Conclusions: The authors present their findings and briefly discuss them to published data. Comments:

  • In line 308: The authors claim that PSA testing lacks specificity in published as well as their own data. This is not clear in the data presented in the current manuscript (Table 1) and not discussed to published data. [major]
  • In line: 317: What are the mechanisms “traditionally” involved in tumor development? The authors should expand on this and relate their findings to this claim. [minor]
  • In line 349: The previous studies comparing agmatine in the three prostate cancer patient subgroups are not cited here. [major]

Author Response

Abstract: This section is overall well-written.

Comments:

The authors should revise English language throughout the manuscript [minor]

Thank you so much for your comments. We revised accordingly

It is unclear what “borderline” disease is and should be clarified in the text. Are there clinical guidelines that define such disease state in prostate cancer? What are the parameters? [major]

  • We have removed this term and replaced by “Precancerous lesion patients”, to be clearer.  Precancerous lesion patients include two categories: Prostate intraepithelial neoplasia (PIN), particularly high-grade PIN (HGPIN), and atypical small acinar proliferation (ASAP). These classes have been identified as precancerous lesions of the prostate, i.e., precursor lesions of prostate carcinoma. PIN refers to the precancerous end of a morphological spectrum involving cell proliferation in prostatic ducts, ducts, and acini. ASAP differs from HGPIN; HGPIN provides a high predictive value for the future development of adenocarcinoma1 while ASAP has the potential significance of synchronous malignant disease located near the biopsy origin2, both being of great clinical importance for the early diagnosis of PCa.

Introduction: This section presents background information on the prostate cancer screening, including PSA measurement and other diagnostic tests. Comments:

What does “clinically-silent” and “biologically non-aggressive” mean? Do the authors mean clinically undetectable disease? The terminology used is parts of the manuscript is confusing and requires clarification. [major]

Thank you so much for your precise observation and valuable comment.

We added the following details:

Line 49: “Prostate cancer is the silent man’s disease, most cancers arise in the periphery of the prostate gland, and cause symptoms only when they have grown to compress the urethra or invade the sphincter or neurovascular bundle3-9.  The stage of the cancer is helpful in understanding the associated symptoms (if any). Four staging systems have been described, and the tumour, node, metastasis (TNM) system is the most widely used.18 In the TNM system, T1 tumours are defined as clinically silent10.

Line 63: “Prostate cancer is a common disease that affects men, usually in middle age or later. In this disorder, certain cells in the prostate undergo an abnormal proliferation leading to a cancerous transformation. The prostate is a gland that surrounds the male urethra and helps producing semen, the fluid that carries sperm. Early prostate cancer usually does not cause pain, and most affected men exhibit no noticeable symptoms. The diagnosis of PC results from a health screenings of prostate specific antigen (PSA) or a digital rectal exam (DRE). As the tumour grows, signs and symptoms can include difficult starting or stopping the flow of urine, a feeling of not being able to empty the bladder completely, blood in the urine or semen, or pain with ejaculation. However, these symptoms can also occur during other genitourinary conditions. Having one or more of these symptoms does not necessarily mean that a man has prostate cancer. On the other hand, many patients do not exhibit any symptoms, from here on indicated as “clinically silent” (This term is commonly used to indicate this category of patients)

Line 56: “The severity and outcome of prostate cancer exhibits many differences. Early-stage prostate cancer can usually be treated successfully, and some older men have prostate tumours that grow so slowly, biological non-aggressive, that they may never cause health problems during their lifetime, even without treatment. However, in other men, the cancer is much more aggressive; in these cases, prostate cancer can be life-threatening. Some cancerous tumours can invade surrounding tissue and spread to other parts of the body. Tumours that begin at one site and then spread to other areas of the body are called metastatic cancers. The signs and symptoms of metastatic cancer depend on where the disease has spread. If prostate cancer spreads, cancerous cells most often appear in the lymph nodes, bones, lungs, liver, or brain. 

In line 77: Arginine metabolism may have been suggested a therapeutically targetable vulnerability in prostate cancer but claims of “Achilles’ heel’, may be a bit farfetched without solid experimental evidence. [minor]

We modified the text adding also other references to strengthen our hypothesis.

Moreover MD Patil et al. previously published: “arginine dependence of the tumour cells has been considered as the ‘Achilles’ heel”11.

Different Authors evaluated the inability of cancer cells to proliferate in the absence of arginine for their “selective destruction”12. Large numbers of enzyme-based anti-cancer therapies are currently undergoing clinical evaluation. It is encouraging that arginase already have achieved considerable success, without causing detrimental side effects and with high tolerability13, 14.

In line 84: The sentence does not make sense. Consider revising. [minor]

Thank you so much for your suggestion. Yes, we rewrote the sentence to be clearer: “the aim of this study is to investigate the levels of circulating polyamines in a population of subjects with suspected prostate cancer and to understand if these biomarkers could be used as a support for clinicians allowing the discrimination among patients with different outcomes.

Benign prostatic hyperplasia (BPH) and pre-cancerous lesion, including prostatic intraepithelial neoplasia (PIN), are interchangeable terms. By which criteria did the authors distinguish one from the other? They claim the presence of three distinct histopathological groups and base all their analyses on these, but it is not clear how these subgroups were defined. [major]

Thank you so much for your comment. We have addressed it during the revision of the manuscript:

WE incorporated the following details in the introduction section (Line 129)

“BPH and PIN are not interchangeable terms. Benign prostatic hyperplasia (BPH) — also called prostate gland enlargement — is a common condition as men get older. Having BPH does not increase your risk for prostate cancer. An enlarged prostate gland can cause uncomfortable urinary symptoms, such as blocking the flow of urine out of the bladder. On the other hand, PIN is defined as " is a condition defined by neoplastic growth of epithelial cells within pre-existing benign prostatic acini or ducts." Prostatic intraepithelial neoplasia (PIN), particularly high-grade PIN (HGPIN), and atypical small acinar proliferation (ASAP) have been identified as precancerous lesions of the prostate15, 16;  that is, precursor lesions to prostatic carcinoma. The latter are categories at increased risk of developing prostate adenocarcinoma. The chance that a patient diagnosed with ASAP will undergo a cancer is about 40% 17. The morphological appearance of HGPIN (ie, tufting, micropapillary, cribriform, flat) does not necessary correlate with the appearance of cancer, and close clinical follow-up is indicated18. HGPIN yields a high predictive value for future development of adenocarcinoma19 and ASAP has the potential significance of synchronic malignant disease located near the origin of the biopsy 20, both are of great clinical importance regarding early diagnosis of PCa.

Therefore, all patients who have been diagnosed with ASAP should repeat biopsies within a period of three to six months.  Whether or not the extent of high-grade PIN in biopsies is a predictor of subsequent prostate cancer is still controversial21.

What does ASAP stand for? [major]

As described in the text (Line 137)

“Atypical small acinar proliferation (ASAP) has been identified as precancerous lesions of the prostate, is a descriptive terminology used in the pathology report of a needle prostate biopsy. The term ASAP was coined by Iczkowski et al in 1997. At a consensus meeting in 2004 sponsored by the World Health Organisation 21, committee members recommended that ASAP should be designated as suspected or highly suspected for cancer15.

Materials and Methods: The experimental procedures and study design are comprehensively explained apart from the precise and clear definition of the histopathological patient groups. In some cases, the authors refer to them as PC, PL and IBPH (Table 1), and others as YES, BORDER and NO (Table 2). What is the difference between hypertrophy and heteroplasia? Why is the distinction between these two subgroups important? [major]

We consider as "bordeline" patients with precancerous lesion. This term is now replaced by the term “Precancerous lesion”. Below we specify the difference between hypertrophy and heteroplasia.

Heteroplasia is an abnormality in the differentiation process of a tissue. The term is used to refer to tumor proliferative processes, which are instead correctly referred to as neoplasia. Hypertrophy is the increase in the volume of a given tissue or organ. It does not include an increase due to the development of adhesions or accumulation of fat, or due to the proliferation of cells. Hypertrophy is due only to the enlargement of the cells of the given tissue or organ. It occurs in permanent cells (non-dividing)22.

We have eliminated the term “heteroplasia” from the text to avoid confusion.

Results and Figures: The results and figures are severely lacking. Comments:

The authors do not describe their results. Rather they outline what each figure shows without any context for the reader. The manuscript would benefit from better organization and potentially subheading in the result section. [major]

We reorganized the paragraph accordingly.

Table 1: PSA levels in the PL group are lower than of IBPH, but still there is not significant difference compared to PC. Could the authors clarify as to why that may be? [minor]

Analysis of the correlation between diagnosis and PSA value shows a statistically significant negative correlation, meaning that PSA value is higher in patients with cancer, and lower in patients with benign prostatic hyperplasia. PSA increases with age, the incidence of cancer increases with age, and it also increases with prostate volume. Considering inconsistencies in PSA measurement, the only diagnostic tool that can confirm cancer is histopathological analysis of biopsy material. Since PSA is organ specific and not tumour specific, the release increases with gland enlargement which does not always correspond to the presence of precancerous lesions and vice versa.

Figure 1 is not very informative and may be redundant. The author’s effort to showcase metabolic pathways of the detected polyamine is appreciated but the presentation is lacking. [minor].

We eliminated figure 1.

Figures 2 and 3 could potentially be merged or figure 2 omitted from the main text. They show the same point. Also, the font size is too small to read the figures. [minor]

Figures 2 and 3 are different modelling methods for multivariate analysis (see below). Both of them are needed to strengthen our results.

O-PLS-DA uses the same basic statistic but uses a mathematical filter to remove systematic variance in the dataset that is unrelated to the sample class (or Y dummy variable). The orthogonal matrix is simpler to explore the 'orthogonal' components easily to fully and understand data set. The characteristics of the OPLS method have been investigated for the purpose of discriminant analysis (OPLS-DA), demonstrating how class-orthogonal variation can be exploited to augment classification performance in cases where the individual classes exhibit divergence in within-class variation. This enables the use of the class-orthogonal variation in a proper statistical context23. Figure 2 does not exclude figure 3, so we leave them both.

In line 250: The authors jump straight to Figure 6, bypassing Figures 4 and 5. [major]

We modified accordingly

Figures 4 and 5 are extremely hard to read and is unclear what in the comparisons are and to what extent. Are the authors comparing different conditions, the expression of different polyamines? [major]

In figure 4 and 5 we describe “Variable importance “in orthogonal projections to latent structures (OPLS). These figures show variable influence on projection (VIP). VIP is a parameter used for calculating the cumulative measure of the influence of individual X-variables on the model. Variable influence on projection is applied in multivariate clinical data analysis to achieve an improved diagnosis of process dynamics.  Variable influence on projection (VIP) is commonly used to summarize the importance of the X-variables in multivariate models based on projections to latent structures i.e., PLS and OPLS methods. VIP values as or larger than 1 point to the most relevant variables, and generally VIP values below 0.5 are considered irrelevant variables.

Thus, VIP for OPLS with positive contribution scores correspond to the metabolites contributing to class discrimination in the PLS-DA model. The number of terms in the sum depends on the number of PLS-DA components that are significant for distinguishing classes. The Y-axis shows the VIP scores corresponding to each variable on the X-axis.

Multivariate analysis based on partial least squares (i.e., PLS and orthogonal projections to latent structures [OPLS]) models has become a useful and appreciated toolbox in clinical research. Univariate analysis is the most basic form of statistical data analysis technique. When the data contains only one variable and doesn’t deal with a causes or effect relationships then a Univariate analysis technique is used. Multivariate analysis is a more complex form of statistical analysis technique and used when there are more than two variables in the data set, as in clinical data, to understand the relationship of each variable with each other.  Most multivariate analysis involves a dependent variable and multiple independent variables. Most univariate analysis emphasizes description while multivariate methods emphasize hypothesis testing and explanation. Perhaps the greatest similarity between univariate and multivariate statistical techniques is that both are important for understanding and analyzing extensive statistical data. Univariate analysis acts as a precursor to multivariate analysis and that a knowledge of the former is necessary for understanding the latter. To avoid model overfitting, the corresponding PLS-DA models were validated by 300-fold permutation tests, shown in Figure 6. The resulting regression lines showed an intercept of R2 at 0.0249 and an intercept of Q2 at -0.246, indicating a valid model. This difference was maintained when the paired analysis of the samples from three groups was split. By OPLS-DA there was a statistically significant difference between the three classes of patients. The scoring plot of the predictive component showed no overlap.

In Figure 6, what to the number 1, 2, 3 stand for? [minor]

We specified as requested

Figure 8 is the most clearly presented piece of data in the manuscript. It shows that Arginine decarboxylase (ADC) expression is significantly different between PC and either PL or IBPH. This point is not discussed in the manuscript as it suggests that this assay may be as diagnostically powerful as the PSA screening (Table 1). [major]

We have modified the manuscript discussing this point.

Line “ROC curves were used to quantify the predictive accuracy of the metabolites. Analysis of the agmatine panel showed an AUC of 0.959 and P ≤ 0.001 in classifyng patients into three gropus (Figure 5). Combining the variables of the three groups with the analyte panel yielded an AUC of 0.948, P ≤ 0.001, demonstrating the robust performance of the panel combined together (Figure 7).  To confirm the results obtained in the measurement of agmatine, we quantified arginine decarboxylase.

Lack of additional PSA – polyamine expression correlation analyses that would enhance the clinical relevance of the study [major]

Other curved ROCs of other polyamines have been added in the supplementary materials

Discussion and Conclusions: The authors present their findings and briefly discuss them to published data. Comments:

In line 308: The authors claim that PSA testing lacks specificity in published as well as their own data. This is not clear in the data presented in the current manuscript (Table 1) and not discussed to published data. [major]

We clarified that agmatine can support PSA for diagnosis and differentiation of patients.

In line: 317: What are the mechanisms “traditionally” involved in tumor development? The authors should expand on this and relate their findings to this claim. [minor]

We have added the mechanisms “traditionally” involved in tumor development.

In line 349: The previous studies comparing agmatine in the three prostate cancer patient subgroups are not cited here. [major]

Agmatine and its plasma levels have never been studied in patients with prostate cancer and these three classes.

Reference:

  1. Bostwick DG, Qian J: High-grade prostatic intraepithelial neoplasia. Modern pathology 2004, 17:360-379.
  2. Warlick C, Feia K, Tomasini J, Iwamoto C, Lindgren B, Risk M: Rate of Gleason 7 or higher prostate cancer on repeat biopsy after a diagnosis of atypical small acinar proliferation. Prostate cancer and prostatic diseases 2015, 18:255-259.
  3. Horwich A, Waxman J, Abel P: The Oxford textbook of oncology. 2001.
  4. Ornstein DK, Rao GS, Smith DS, Andriole GL: The impact of systematic prostate biopsy on prostate cancer incidence in men with symptomatic benign prostatic hyperplasia undergoing transurethral resection of the prostate. The Journal of urology 1997, 157:880-884.
  5. Merrill RM, Feuer EJ, Warren JL, Schussler N, Stephenson RA: Role of transurethral resection of the prostate in population-based prostate cancer incidence rates. American journal of epidemiology 1999, 150:848-860.
  6. Shibata A, Ma J, Whittemore AS: Prostate cancer incidence and mortality in the United States and the United Kingdom. JNCI: Journal of the National Cancer Institute 1998, 90:1230-1231.
  7. Gann PH, Hennekens CH, Stampfer MJ: A prospective evaluation of plasma prostate-specific antigen for detection of prostatic cancer. Jama 1995, 273:289-294.
  8. Potosky AL, Feuer EJ, Levin DL: Impact of screening on incidence and mortality of prostate cancer in the United States. Epidemiologic reviews 2001, 23:181-186.
  9. Grönberg H: Prostate cancer epidemiology. The Lancet 2003, 361:859-864.
  10. Hamilton W, Sharp D: Symptomatic diagnosis of prostate cancer in primary care: a structured review. British Journal of General Practice 2004, 54:617-621.
  11. Patil MD, Bhaumik J, Babykutty S, Banerjee UC, Fukumura D: Arginine dependence of tumor cells: targeting a chink in cancer’s armor. Oncogene 2016, 35:4957-4972.
  12. Butler M, van der Meer LT, van Leeuwen FN: Amino acid depletion therapies: starving cancer cells to death. Trends in Endocrinology & Metabolism 2021, 32:367-381.
  13. Yau T, Cheng P, Chan P, Chan W, Chen L, Yuen J, Pang R, Fan S, Poon RT: A phase 1 dose-escalating study of pegylated recombinant human arginase 1 (Peg-rhArg1) in patients with advanced hepatocellular carcinoma. Investigational new drugs 2013, 31:99-107.
  14. Feun L, You M, Wu C, Kuo M, Wangpaichitr M, Spector S, Savaraj N: Arginine deprivation as a targeted therapy for cancer. Current pharmaceutical design 2008, 14:1049-1057.
  15. Koca O, Çalışkan S, Öztürk Mİ, Güneş M, Karaman MI: Significance of atypical small acinar proliferation and high-grade prostatic intraepithelial neoplasia in prostate biopsy. Korean Journal of Urology 2011, 52:736-740.
  16. Srirangam V, Rai BP, Abroaf A, Agarwal S, Tadtayev S, Foley C, Lane T, Adshead J, Vasdev N: Atypical small acinar proliferation and high grade prostatic intraepithelial neoplasia: should we be concerned? An observational cohort study with a minimum follow-up of 3 years. Current Urology 2016, 10:199-205.
  17. Scattoni V, Roscigno M, Freschi M, Dehò F, Raber M, Briganti A, Fantini G, Nava L, Montorsi F, Rigatti P: Atypical small acinar proliferation (ASAP) on extended prostatic biopsies: predictive factors of cancer detection on repeat biopsies. Archivio Italiano di Urologia, Andrologia: Organo Ufficiale [di] Societa Italiana di Ecografia Urologica e Nefrologica 2005, 77:31-36.
  18. Bishara T, Ramnani DM, Epstein JI: High-grade prostatic intraepithelial neoplasia on needle biopsy: risk of cancer on repeat biopsy related to number of involved cores and morphologic pattern. The American journal of surgical pathology 2004, 28:629-633.
  19. Klink JC, Miocinovic R, Galluzzi CM, Klein EA: High-grade prostatic intraepithelial neoplasia. Korean journal of urology 2012, 53:297-303.
  20. Andras I, Telecan T, Crisan D, Cata E, Kadula P, Andras D, Bungardean M, Coman I, Crisan N: Different clinical significance of ASAP/HGPIN pattern in systematic vs. MRI‑US fusion guided prostate biopsy. Experimental and Therapeutic Medicine 2020, 20:1-1.
  21. Epstein JI, Potter SR: The pathological interpretation and significance of prostate needle biopsy findings: implications and current controversies. The Journal of urology 2001, 166:402-410.
  22. Baba AI, Câtoi C: Comparative oncology: Publishing House of the Romanian Academy Bucharest, 2007.
  23. Bylesjö M, Rantalainen M, Cloarec O, Nicholson JK, Holmes E, Trygg J: OPLS discriminant analysis: combining the strengths of PLS‐DA and SIMCA classification. Journal of Chemometrics: A Journal of the Chemometrics Society 2006, 20:341-351.

Round 2

Reviewer 2 Report

The authors have addressed the majority of the concerns raised and the manuscript has significantly improved. English language has been corrected and the overall presentation has improved. The author’s attempt to comprehensively address the issues raised is greatly appreciated. One minor note is to consider rephrasing the sentence in line 49: “silent man’s disease”, as it doesn’t make sense to this reader.